# Ranking Items by the Current-Preferences and Profits: A List-wise Learning-to-Rank Approach to Profit Maximization

## Abstract

In e-commerce platforms, *profit-aware recommender systems* aim to *improve the platform's profits while maintaining high overall accuracy* by recommending items with high profits as *top-ranked* items. We explore two issues faced by existing *model-based profit-aware approaches* (*i.e.*, MBAs) when training recommendation models for profit enhancement. First, current MBAs tend to *inaccurately* infer the item ranking by the *profit-based weighting scheme*; the ranking of observed (*i.e.*, purchased) items by a user is inferred *without considering the user preference* for each item, while all unobserved items are assumed to have an *equally low ranking*. Second, current MBAs train the model *without employing* the item ranking as *ground truth*; during training, the model is optimized for the preference score for each item *independently* rather than *being directly optimized for the overall ranking of items*. To tackle these issues, we propose a novel MBA that involves three key steps: **(S1)** defining the *Current Preference incorporated with Profit* (*i.e.*, CPP) for items; **(S2)** classifying items through CPP; and **(S3)** training the model by *list-wise learning-to-rank* (LTR) based on CPP. Extensive experimental results using real-world platform datasets demonstrate that our approach improves accuracy by approximately 4% and profits by about 24% compared to the best-competing method.

## CCS Concepts

• **Information systems** → **Recommender systems**.

## Keywords

Collaborative filtering, list-wise learning-to-rank, profit maximization, recommender systems

**ACM Reference Format:**
Anonymous Author(s). 2025. Ranking Items by the Current-Preferences and Profits: A List-wise Learning-to-Rank Approach to Profit Maximization. In *Proceedings of Make sure to enter the correct conference title from your rights confirmation email (WWW'25)*. ACM, New York, NY, USA, 12 pages. https://doi.org/XXXXXXX.XXXXXXX

## 1 Introduction

Typically, in such domains as e-commerce, when a user purchases an item, the platform (*e.g.*, Amazon and eBay) gains a certain amount of *profit* from that item. The profit may vary from item to item (*i.e.*, items are *not equally profitable*). For example, the *products*

*with higher margins* contribute more to platforms' profit enhancement when purchased by users. In this context, several studies have been conducted to develop *profit-aware recommender systems*, which pursue the following two goals [1, 4, 8, 18, 21, 23, 26, 35]: **(G1)** *maintaining the overall accuracy* of recommendations compared to baseline models that do not consider item profits; and **(G2)** *increasing the platform's profits* by expanding recommendations of *items with high profits* (*i.e.*, *profitable items*). To achieve these two goals, there have been a number of attempts to develop profit-aware recommendation methods, which are classified into two groups depending on when the step for profit maximization is accomplished: (i) the *re-ranking-based approach* (namely, RBA) [1, 8, 18, 23, 26] and (ii) the *model-based approach* (namely, MBA) [4, 21, 35].

The RBA (*e.g.*, CPP-PPR [18] and Rec-RL [26]) trains the recommendation model by using the same loss function as that of the baseline model; the *cross-entropy loss* [2, 3, 14] is minimized in the sense that the preference scores for *observed* (*i.e.*, purchased) items of each user approach 1 while the scores for *unobserved* (*i.e.*, non-purchased) items approach 0 [10, 11, 13]. After completing the model training, the RBA *combines* the preference scores predicted by the model with item profits. Then, the RBA *re-ranks* the user's unobserved items so that the items with not only high preference scores but also high profits are ranked *high*. By recommending items placed in the top ranks to her, the RBA aims to achieve both **(G1)** and **(G2)**.

On the other hand, the MBA (*e.g.*, PE-LTR [21] and VCF [4]), which has been more actively researched recently, attempts to achieve both **(G1)** and **(G2)** *during the model training stage*. For example, PE-LTR [21] trains a model for **(G1)** and **(G2)** by using respective loss functions: (i) the loss function for **(G1)** is the one originally used in the baseline model (*i.e.*, cross-entropy); (ii) the loss function for **(G2)** exploits the *profit* from each observed item as a *weight* to train the model to predict *higher preference scores for the items with higher profits* (*i.e.*, the *profit-based weighting scheme*); and (iii) the model is trained to find a *Pareto-efficient solution* [6, 30, 39] between these two loss functions, thereby mainly recommending profitable items that she would prefer.

In this paper, we start by performing a comprehensive analysis for prior work on the MBA. First, our analysis reveals that existing MBAs [4, 21, 35] have the following *two issues* when it comes to model training for **(G2)**:

- **(I1)** Inferring item rankings *regardless of user preference* through the *profit-based weighting scheme*
- **(I2)** Model training by *point-wise learning-to-rank* (LTR) [15, 24] that is *not directly optimized for the item ranking*

In **(I1)**, the profit-based weighting scheme is designed to train the model to infer higher preferences for the observed items with higher profits. However, since users on the e-commerce platforms are typically *unaware of* the item profits (*i.e.*, item profits are *not visible* to users), their selections have been made *independently of profit*

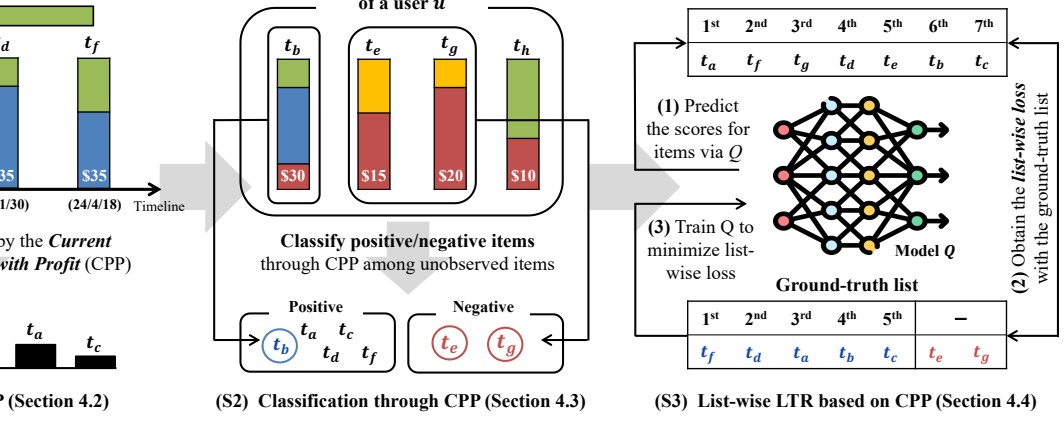

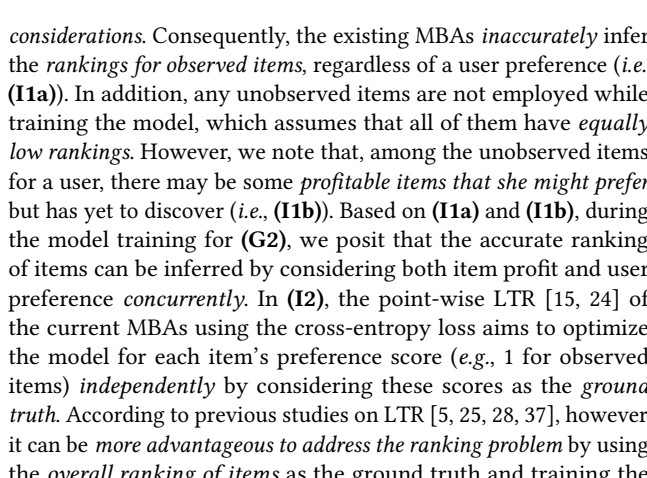

**Figure 1: Overview of our proposed approach. Colored (*e.g.*, blue or green) bars represent items' latent attributes, the date at the bottom shows when a user used items, and the symbol $ represents the items' profit.**

*considerations*. Consequently, the existing MBAs *inaccurately* infer the *rankings for observed items*, regardless of a user preference (*i.e.*, **(I1a)**). In addition, any unobserved items are not employed while training the model, which assumes that all of them have *equally low rankings*. However, we note that, among the unobserved items for a user, there may be some *profitable items that she might prefer* but has yet to discover (*i.e.*, **(I1b)**). Based on **(I1a)** and **(I1b)**, during the model training for **(G2)**, we posit that the accurate ranking of items can be inferred by considering both item profit and user preference *concurrently*. In **(I2)**, the point-wise LTR [15, 24] of the current MBAs using the cross-entropy loss aims to optimize the model for each item's preference score (*e.g.*, 1 for observed items) *independently* by considering these scores as the *ground truth*. According to previous studies on LTR [5, 25, 28, 37], however, it can be *more advantageous to address the ranking problem* by using the *overall ranking of items* as the ground truth and training the model to be *directly optimized for this ranking*.

To address these issues, we propose a novel MBA with three key steps; the schematic overview is illustrated in Figure 1 with a user $u$ and eight items (*i.e.*, $t_a$, $t_b$, ..., and $t_h$).

**(S1) Definition of CPP.** To tackle **(I1a)**, we determine the *current preference* (CP) for each observed item by evaluating (i) the degree to which the item has attributes the user prefers and (ii) when the user has purchased the item. By using the *CP incorporated with Profit* (referred to as CPP) for each item, we can effectively rank the observed items. In **(S1)** of Figure 1, among the observed items by $u$ ($t_a$, $t_c$, $t_d$, and $t_f$), both items $t_d$ and $t_f$ have a *higher degree of* having preferred attributes (*i.e.*, attributes marked in blue or green) and have been used *more recently* than $t_a$ and $t_c$; thus, they are placed in higher ranks than $t_a$ or $t_c$. (Please refer to Section 4.2 for more details on determining the CPP for each observed item.)

**(S2) Classification through CPP.** To tackle **(I1b)**, we infer the CPP for each unobserved item and classify them as having either high or low CPP. We then designate unobserved items with high CPP as the user's *positive items* along with the observed items, and unobserved items with low CPP as the user's *negative items*. In **(S2)**

of Figure 1, among the unobserved items by $u$, we designate $t_b$ as her positive item due to its superior preferred attributes and higher profits compared to the others. (Please refer to Section 4.3 for more details on classifying unobserved items based on CPP.)

**(S3) List-wise LTR based on CPP.** To tackle **(I2)**, we aim to design a novel model training scheme that allows the model to be *directly optimized for the item ranking* determined by the CPP for each item. Specifically, we regard the list of items ranked according to CPP as the *ground-truth list* and the list of items ranked according to the scores predicted by the model as the *predicted list*. By minimizing the *list-wise loss* [5, 27, 37] between these two lists, we train the model to achieve the following objectives: (i) predicting positive items to be *ranked higher* than negative items; and (ii) predicting positive items with higher CPP to be *ranked higher* than those with lower CPP. In **(S3)** of Figure 1, we train the model $Q$ to *minimize the difference* between the ranking list of items predicted by $Q$ (*i.e.*, predicted list) and the ground-truth list. (For more details on composing the ground-truth list, please refer to Section 4.4.)

It is worth noting that our approach is *orthogonal* to any recommendation (*i.e.*, collaborative filtering, CF) model (*e.g.*, MF [24] and LightGCN [12]) (*i.e.*, *model-agnostic*), as existing MBAs.

The technical novelty of this study can be summarized as follows.

- *Key Observation*: We identify the issues (*i.e.*, **(I1)** and **(I2)**) of existing MBAs in terms of training the model for **(G2)**.
- *Novel Approach*: We introduce the CPP for *accurate inference* of the item ranking, thereby training the model to be *directly optimized* for the list of items ranked by the CPP.
- *Extensive Evaluation*: We empirically verify that our approach is more advantageous in *improving both profit and accuracy* than existing MBAs over various real-world datasets.

## 2 Preliminaries

### 2.1 Problem Definition

Let $\mathcal{U}$ and $\mathcal{T}$ denote the sets of all users and all items, respectively. When a user $u \in \mathcal{U}$ uses (*i.e.*, purchases) an item $t \in \mathcal{T}$, we denote

the *unique profit* obtained from $t$ as $prof(t)$. For each user $u$, the set of all *observed* (*i.e.*, purchased) items and the set of all *unobserved* (*i.e.*, non-purchased) items are denoted as $\mathcal{T}_u$ and $\overline{\mathcal{T}}_u$, respectively. Given the set $Rec_u(N)$ of top-$N$ items for $u$ recommended by the model $Q$ and the set $Rel_u$ of *relevant* items (*i.e.*, ground truth) for $u$, the problem of *item recommendation for profit maximization* is formally defined to achieve the following goals:

- **(G1)** Recommending with *high accuracy* by including as many items belonging to $Rel_u$ as possible in $Rec_u(N)$;
- **(G2)** Achieving a platform's *high profit* by including more *items with high profits* (*i.e.*, profitable items) in $Rec_u(N)$ than others.

Notations used in this paper are summarized in Appendix A.

## 2.2 Existing Approaches

The existing profit-aware recommendation methods are divided into two groups based on when the step for profit maximization is accomplished: (i) RBA [1, 8, 18, 23, 26]; (ii) MBA [4, 21, 35].

**Re-ranking-based approach (RBA).** The RBA (*e.g.*, CPP-PPR [18] and Rec-RL [26]) trains the recommendation model by using the *original loss function* (*e.g.*, the cross-entropy loss function), which does *not* consider the item profit, shown below:

$$\mathcal{L}_u^{acc} = -\left( \sum_{t \in \mathcal{T}_u} log(\hat{Q}(u,t)) + \sum_{t \in \overline{\mathcal{T}}_u} log(1 - \hat{Q}(u,t)) \right), \quad (1)$$

where $\hat{Q}(u,t)$ indicates the preference score of $u$ for $t$ predicted by $Q$ (*i.e.*, $0 \le \hat{Q}(u,t) \le 1$). Subsequently, the RBA determines the *re-ranking score* of $u$ for $t$, denoted as $RRS(u,t)$, by combining $\hat{Q}(u,t)$ and $prof(t)$, which can be formally defined as follows [23]:

$$RRS(u,t) = w \cdot \hat{Q}(u,t) + (1-w) \cdot prof(t), \quad (2)$$

where $w$ ($0 \le w \le 1$) represents the balancing parameter for $\hat{Q}(u,t)$ and $prof(t)$. As $w$ decreases, the profitable items can be recommended to *more users*, even including those who are predicted to have *low preference scores* for profitable items; *i.e.*, **(G2)** can be achieved, but **(G1)** may not be achieved. On the other hand, as $w$ increases, items predicted to have *high preference scores* from a user will be mainly recommended *regardless of the profits*; *i.e.*, **(G1)** can be achieved, but **(G2)** may not be. To balance between the two goals, existing RBAs (i) determine $w$ *experimentally* by regarding it as a hyperparameter [18, 23] or (ii) utilize deep-learning techniques, such as *reinforcement learning* [26], to find a suitable value for $w$.

**Model-based approach (MBA).** The MBA (*e.g.*, PE-LTR [21]) employs the following loss function for the model training:

$$\mathcal{L}_u = w_1 \cdot \mathcal{L}_u^{acc} + w_2 \cdot \mathcal{L}_u^{prof}, \quad (3)$$

where $\mathcal{L}_u^{acc}$ denotes the loss function in Eq. (1) to achieve **(G1)**; $\mathcal{L}_u^{prof}$ denotes the loss function to achieve **(G2)**; and $w_1$ and $w_2$ represent the balancing parameters for $\mathcal{L}_u^{acc}$ and $\mathcal{L}_u^{prof}$, respectively. To compute $\mathcal{L}_u^{prof}$, the MBA applies the *profit-based weighting scheme* for the cross-entropy loss as follows:

$$\mathcal{L}_u^{prof} = -\left( \sum_{t \in \mathcal{T}_u} h(prof(t)) \cdot log(\hat{Q}(u,t)) \right), \quad (4)$$

where $h(\cdot)$ denotes the monotonically *non-decreasing* (*e.g.*, logarithmic) function for the profit $prof(t)$ from the observed item $t$ by $u$. In other words, $h(prof(t))$ can be referred to as the *weight* enabling $t$ with higher $prof(t)$ to be learned as a *more-confident positive* item for $u$. Thus, $\mathcal{L}_u^{prof}$ decreases, as profitable items are predicted to have higher preference scores by $Q$; *i.e.*, the optimization by $\mathcal{L}_u^{prof}$ can let $Q$ recommend profitable items to *more users* than the baseline without $\mathcal{L}_u^{prof}$. Finally, the MBA aims to achieve *both* goals by training $Q$ in a direction where both $\mathcal{L}_u^{acc}$ and $\mathcal{L}_u^{prof}$ are optimized, along with finely tuned $w_1$ and $w_2$ (*e.g.*, *Pareto-efficient solutions* [6, 30, 39] by PE-LTR [21]).

For details on each method within the RBA and MBA, please refer to Appendix D.

## 3 Motivation: Limitations of the Current MBAs

We begin by revisiting the following issues of existing MBAs with respect to training the model for **(G2)**: **(I1)** *inaccurate inference for the item ranking*; and **(I2)** *model training that is not directly optimized for the item ranking*. (Note that, in this section, we mainly focus on an in-depth analysis of **(I1)**; the empirical evaluation to handle **(I2)** is presented in Section 5.) First, we highlight that the *profit-based weighting scheme* in Eq. (4) trains the model to infer preferences for observed items in the order of their profits. However, it can train the model to *inaccurately predict the user preferences* for observed items, leading to a *decrease* in recommendation accuracy. This is because, typically, an item profit is *not revealed* to a user; thus, she selects items entirely *independently of the item profit*. Consequently, the model may be trained to infer the ranking for observed items in the order *unrelated to her actual interest* (*i.e.*, **(I1a)**). Furthermore, it is worth noting that existing MBAs do *not consider any unobserved items* while training the model; *i.e.*, all unobserved items are assumed to have *equally low rankings*. However, it is important to recognize there could be unobserved items that the user would prefer but has not come across yet (*i.e.*, **(I1b)**).

To demonstrate these issues, we empirically compared the accuracy and profit of the MBA with those of RBA by employing Spons-Rec [23] and PE-LTR [21] for the benchmark RBA and MBA, respectively. Note that the RBA does not perform *separate* model training for profit enhancement (*i.e.*, **(G2)**); it only combines (*i.e.*, *post-processes*) the preference scores from the model trained only for **(G1)** with the item profit to determine top-$N$ items. In this sense, if the MBA has *lower accuracy or profit* than that of the RBA, then it verifies the *limitations* of the profit-based weighting scheme employed in the MBA to achieve **(G2)**. We adopted *Recall* (*i.e.*, $R@N$) and *Total Profit* (*i.e.*, $TP@N$) to evaluate accuracy and profit from top-$N$ recommendations, where $N=10$. (Please refer to Appendix B for the details of metrics.) As recommendation models, we employed MF [24], CDAE [36], and LightGCN [12].

In Table 1, for both MF and CDAE, the MBA's accuracy is *even lower* than that of the RBA (*i.e.*, about -0.9% and -1.1% for MF and CDAE, respectively, compared to the RBA). In addition, for both MF and LightGCN, the MBA shows *lower profit* than the RBA by about -0.5% and -1.3%, respectively. These results indicate the MBA fails to achieve the two goals **(G1)** and **(G2)**. In the following section, we introduce our novel MBA to profit maximization, which successfully addresses these issues of current MBAs.

**Table 1: Comparison of accuracy and profit between RBA and MBA, where 'Baseline' denotes the corresponding CF model that does not consider item profit. The underlined values indicate better results among those of RBA and MBA (ABeauty).**

| Methods | Accuracy (*i.e.*, R@10) | | | Profit (*i.e.*, TP@10) | | |
|---|---|---|---|---|---|---|
| | **MF** | **CDAE** | **LightGCN** | **MF** | **CDAE** | **LightGCN** |
| **Baseline** | 0.0651 | 0.0670 | 0.0688 | 27.2130 | 28.3342 | 30.2029 |
| **RBA** | 0.0655 | 0.0662 | 0.0680 | 29.0829 | 29.8063 | 30.4253 |
| **MBA** | 0.0649 | 0.0655 | 0.0683 | 28.9407 | 29.9093 | 30.0188 |

## 4 Our Proposed Approach

### 4.1 Overview

Our proposed MBA trains the recommendation model (*e.g.*, MF [24], CDAE [36], LightGCN [12]) to satisfy goals **(G1)** and **(G2)** with the following three key steps:

- **(S1)** Definition of CPP: We define and compute the CPP to infer the difference in preferences among the observed items by a user, thereby *ranking them accurately*.
- **(S2)** Classification through CPP: We classify unobserved items as the user's positive or negative items based on the CPP for each item.
- **(S3)** List-wise LTR based on CPP: Using the list-wise LTR techniques [5, 27, 37], we train the model to be *directly* optimized for the list of items *ranked based on CPP*.

We elaborate on **(S1)**, **(S2)**, and **(S3)** in the following Sections 4.2, 4.3, and 4.4, respectively. The schematic overview of our approach is depicted in Figure 1.

### 4.2 Definition of CPP (S1)

In the e-commerce domain, even for the items $\in \mathcal{T}_u$ that a user $u$ has used (*i.e.*, purchased), the degree to which they are *close to her current interests* could be different. To identify these differences among observed items by $u$, we infer the *current preference* (CP) for each observed item based on the following two factors: (i) the degree to which the item has attributes she prefers and (ii) when she has used the item. For (i), to infer the degree of having the attributes preferred by $u$, we employ a (distinct) model $Q'$, which has been pre-trained *in advance* following Eq. (1); *i.e.*, if $t$ has a high (resp. low) score predicted by $Q'$, then we can recognize $t$ as an item with a high (resp. low) degree of having attributes that $u$ prefers.[1] For (ii), to distinguish between items that $u$ has *recently or previously* used, we consider the *chronological order* in which each item $t$ was used among $\mathcal{T}_u$. Following the intuition from existing e-commerce research [9, 29], we assume that recently purchased items may *better align with* the user's current interests than previously purchased items.

In this context, CP of $u$ for $t$ (*i.e.*, $CP(u, t)$) can be formulated as follows:

$$CP(u, t) = \hat{Q}'(u, t) \cdot recency(u, t), \quad t \in \mathcal{T}_u. \tag{5}$$

---

[1]We also empirically verified the results using the *dynamic* preference scores by the model being (currently) trained and observed a similar trend when using $Q'$. In this paper, we thus use the pre-trained model $Q'$ for the efficiency of the model training.

where $\hat{Q}'(u, t)$ represents the preference score of $u$ predicted by the pre-trained model $Q'$ for $t$; and $recency(u, t)$ represents a score indicating how recently $u$ has used $t$, which is formulated as follows:

$$recency(u, t) = \log(seq(u, t) + 1), \quad t \in \mathcal{T}_u, \tag{6}$$

where $seq(u, t)$ denotes the chronological order in which each item $t$ has been used by $u$ (*i.e.*, $1 \leq seq(u, t) \leq |\mathcal{T}_u|$). For example, the item $t$ that has been used by $u$ the longest time ago has a value of 1 for $seq(u, t)$. Depending on $seq(u, t)$, the items used *more recently* will have higher $recency(u, t)$, where a *logarithmic function* is used to decrease the score for items used longer ago more significantly.

To address **(I1a)**, we can effectively rank observed items $\in \mathcal{T}_u$ of $u$ according to the *CP incorporated with Profit*, namely CPP, as follows:

$$CPP(u, t) = CP(u, t) \cdot h(prof(t)), \quad t \in \mathcal{T}_u. \tag{7}$$

where $h(\cdot)$ denotes the monotonically *non-decreasing* function for $prof(t)$. We employed a *logarithmic function* for $h(\cdot)$, following the existing study for the MBA [21].

In **(S1)** of Figure 1, among observed items by $u$ (*i.e.*, $t_a$, $t_c$, $t_d$, and $t_f$), items $t_d$ and $t_f$ have a high degree of having attributes that $u$ prefers (*i.e.*, attributes marked in blue or green) and have been purchased more recently than $t_a$ or $t_c$; thus, $t_d$ and $t_f$ can be inferred to have a higher CP than others. Between $t_d$ and $t_f$, we place $t_f$ in the first rank since it has been purchased more recently than $t_d$. Next, for $t_a$ and $t_c$, $t_c$ can be inferred to have a slightly higher CP than $t_a$, but it has a *much smaller profit*. For this reason, we rank $t_c$ lower than $t_a$ by comparing the CPPs of two items.

### 4.3 Classification through CPP (S2)

We note that *not all* unobserved items $\in \overline{\mathcal{T}}_u$ of a user $u$ *uniformly* have a *low* degree of having attributes preferred by $u$; *i.e.*, there may be some items with the attributes that she would prefer, but she has not yet been able to use them because she is *unaware of their existence*. If we identify profitable items among such items and use them along with the observed items for model training, then the model $Q$ might be trained to recommend profitable items to *more users* with high accuracy.

Thus, to address **(I1b)**, we obtain $CP(u, t)$ for each item $t \in \overline{\mathcal{T}}_u$ as in Eq. (5); then, we distinguish the items with a high degree of having preferred attributes from those with a low degree of having preferred attributes. Here, since $t \in \overline{\mathcal{T}}_u$ has not yet been used by $u$, we compute $recency(u, t)$ by regarding $seq(u, t)$ equally as 1. The set $\mathcal{H}_u$ of unobserved items with high CP is formally defined as follows:

$$\mathcal{H}_u = \left\{ t \mid rank_{u,t} \leq |\overline{\mathcal{T}}_u| \times \frac{\alpha}{100}, \ t \in \overline{\mathcal{T}}_u \right\}, \tag{8}$$

where $rank_{u,t}$ denotes the position of $t$ when all items $\in \overline{\mathcal{T}}_u$ are sorted in descending order by $CP(u, t)$; and $\alpha$ denotes a hyperparameter for the proportion of items to be included in $\mathcal{H}_u$. That is, we determine the set of unobserved items of $u$ in the top $\alpha$% based on the $CP(u, t)$ as $\mathcal{H}_u$.

Finally, among items $\in \mathcal{H}_u$, we randomly sample $\beta$% of items with high profits, according to the sampling probability $P(t)$ defined as follows:

$$P(t) \propto f(prof(t)) := prof(t)^2, \tag{9}$$

where $f(\cdot)$ represents the function to compute the sampling probability. (Although various arithmetic functions (*e.g.*, log, squared root, square) can be adopted as $f(\cdot)$, we employed the *square*, which showed the best results.) We consider the items sampled from $\mathcal{H}_u$ following $P(t)$ to have the high CPP and classify them as *positive items* along with the observed items; *i.e.*, all of these items constitute the set $\mathcal{M}_u$, which represents the *entire positive items* of $u$.

On the other hand, the remaining unobserved items not belonging to $\mathcal{H}_u$ (*i.e.*, items $\in (\overline{\mathcal{T}}_u - \mathcal{H}_u)$) can be regarded as items with *low CP*. In this sense, we *randomly sample* items from the set $(\overline{\mathcal{T}}_u - \mathcal{H}_u)$ and define them as the set $\mathcal{N}_u$ of the *negative items* of $u$. Note that, for $t \in \mathcal{N}_u$, we consider $CP(u, t)$ of $u$ to be 0, and thus $CPP(u, t)$ also becomes 0.

In **(S2)** of Figure 1, we infer the degree of having the attributes preferred by $u$ for each unobserved item (*i.e.*, $t_b$, $t_e$, $t_g$, and $t_h$). Both $t_b$ and $t_h$ have a higher degree of having the preferred attributes (*i.e.*, marked in blue or green) than $t_e$ or $t_g$; *i.e.*, $t_b$ and $t_h$, which have a higher CP, belong to $\mathcal{H}_u$. Between two items $\in \mathcal{H}_u$, we classify $t_b$ as a *positive item* of $u$ since it has higher profits, and the remaining items $\in (\overline{\mathcal{T}}_u - \mathcal{H}_u)$ are classified as *negative items* of $u$.

## 4.4 List-wise LTR based on CPP (S3)

Suppose model $Q$ is trained to be *directly optimized for the item ranking based on CPP*; in that case, it can predict that items with higher CPPs will have higher preference scores while successfully distinguishing negative items from positive ones. As a result, $Q$ can accurately recommend profitable items that $u$ is likely to prefer more, enabling it to achieve both **(G1)** and **(G2)**.

To address **(I2)**, we propose a model training scheme based on *list-wise learning-to-rank (LTR)*; *i.e.*, we compose a list of items $\in \mathcal{M}_u \cup \mathcal{N}_u$ by *ranking* them based on $CPP(u, t)$ obtained by **(S1)** and **(S2)** and use it as the *ground-truth list* $GT_u$ for $u$. Specifically, we sort the list of items belonging to $\mathcal{M}_u \cup \mathcal{N}_u$, as follows: (i) for $t \in \mathcal{M}_u$, we place $t$ with a *higher* value of CPP at a *higher* rank than $t$ with a *lower* value of CPP on the list; and (ii) for $t \in \mathcal{N}_u$, we place $t$ at a *lower rank* on the list than any item $\in \mathcal{M}_u$. Note that we randomly assign the rankings among negative items $\in \mathcal{N}_u$ since $CPP(u, t)$ for $t \in \mathcal{N}_u$ is all equal to 0. With $GT_u$, we design the following *list-wise loss* function[2]:

$$\mathcal{L}_u^{prof} = -\log(R(GT_u|Q)),$$

$$\text{where } R(GT_u|Q) = \prod_{i=1}^{|\mathcal{M}_u|} \frac{exp(\hat{Q}(u, GT_u[i]))}{\sum_{k=i}^{|GT_u|} exp(\hat{Q}(u, GT_u[k]))}, \quad (10)$$

where $GT_u[i]$ denotes the $i$-th ranked item in $GT_u$ (*e.g.*, $GT_u[1]$ indicates the item of the first rank, *i.e.*, the item with the highest value of $CPP(u, t)$); and $\hat{Q}(u, GT_u[i])$ denotes the preference score of $u$ predicted by $Q$ for $GT_u[i]$.

Optimizing $\mathcal{L}_u^{prof}$ indicates *decreasing the difference* between (i) the list of items $t \in \mathcal{M}_u \cup \mathcal{N}_u$ sorted by $\hat{Q}(u, t)$ and (ii) $GT_u$. Specifically, it enables the model $Q$ to learn the order of items $\in \mathcal{M}_u \cup \mathcal{N}_u$ in terms of a user preference, as follows:

- For items $\in \mathcal{M}_u$, the loss gets reduced as $\hat{Q}(u, t)$ for $t$ ranked higher in $GT_u$ is predicted to be *higher* than $\hat{Q}(u, t)$ for $t$ ranked

lower in $GT_u$, *i.e.*, the scores predicted by $Q$ support the *rankings* in $GT_u$.
- For the item $t \in \mathcal{N}_u$, the loss gets reduced as $\hat{Q}(u, t)$ is predicted to be *lower* than any items $\in \mathcal{M}_u$; since $CPP(u, t)$ is uniformly 0, the order among items $\in \mathcal{N}_u$ in a preference is *not considered*.

In **(S3)** of Figure 1, the ground-truth list for $u$ consists of positive items and negative items of $u$. Among positive items, those with high CPPs are ranked higher (*e.g.*, $t_f$ and $t_d$ in the first and second ranks, respectively). Additionally, negative items (*i.e.*, $t_e$ and $t_g$) are ranked lower than positive items (*i.e.*, lower than the fifth rank), and the rankings among negative items are not specified. Then, the list-wise loss $\mathcal{L}_u^{prof}$ is computed by the difference between the list predicted by $Q$ and the ground-truth list for $u$. We train $Q$ until it can predict preference scores for the seven items (*i.e.*, $t_a$, $t_b$, $t_c$, $t_d$, $t_e$, $t_f$, and $t_g$) to support the rankings in the ground-truth list for $u$.

Finally, with $\mathcal{L}_u^{prof}$ developed as in Eq. (10), we train the model $Q$ to effectively achieve both **(G1)** and **(G2)**, as follows:

$$\mathcal{L}_u = \mathcal{L}_u^{acc} + \gamma \cdot \mathcal{L}_u^{prof}, \quad (11)$$

where $\mathcal{L}_u^{acc}$ denotes the original loss function as Eq. (1); and $\gamma$ denotes the hyperparameter to adjust how much $\mathcal{L}_u^{prof}$ is reflected in the final loss. As the value of $\gamma$ increases (resp. decreases), the platform's profits can be enhanced (resp. reduced), but the overall accuracy can be reduced (resp. enhanced).

## 4.5 Analysis of Time Complexity

PROPOSITION 4.1. *The computational complexity for model training is linear in* $|\mathcal{U}|$.

For **(S1)**, the time complexity of computing $CP(u, t)$ for items can be $O(|\mathcal{T}_u|)$ since we employ the pre-trained model $Q'$. Next, defining $\mathcal{H}_u$ requires $O(|\overline{\mathcal{T}}_u| \cdot \log(|\overline{\mathcal{T}}_u|))$, the same as the complexity of the *quick sort* [38]. Then, we can determine $\mathcal{M}_u$ with a complexity of $O(|\mathcal{H}_u| \cdot \log(|\mathcal{H}_u|))$. Since $|\mathcal{H}_u|$ is much smaller than $|\overline{\mathcal{T}}_u|$ and defining $\mathcal{N}_u$ requires only the complexity of $O(1)$, the total complexity for **(S2)** can be approximated to $O(|\overline{\mathcal{T}}_u| \cdot \log(|\overline{\mathcal{T}}_u|))$. The complexity of composing $GT_u$ for **(S3)** is $O(|\mathcal{M}_u| \cdot \log(|\mathcal{M}_u|))$ because we only sort items $\in \mathcal{M}_u$. Obtaining $\mathcal{L}_u^{prof}$ by employing $GT_u$ requires $O(|\mathcal{M}_u| \cdot |GT_u|)$, which can be approximated to $O(|GT_u|^2)$. In summary, the complexity of our **(S1)**–**(S3)** for a user $u$ is $O(|\mathcal{T}_u|) + O(|\overline{\mathcal{T}}_u| \cdot \log(|\overline{\mathcal{T}}_u|)) + O(|GT_u|^2)$. Note that **(S1)** and **(S2)** can be performed *independently* of model training. Thus, the final complexity for training $Q$ with all users becomes $O(|\mathcal{U}| \cdot |GT_u|^2)$, which can be approximated to $O(|\mathcal{U}|)$ since $|GT_u|$ is typically much smaller than $|\mathcal{U}|$ and often does not scale with $|\mathcal{U}|$.

In the **(EQ5)** of the following section, we empirically validate our theoretical findings.

## 5 Empirical Evaluation

### 5.1 Experimental Setup

**Datasets.** We employed three real-world datasets widely used in profit-aware recommender systems [4, 18, 22]: Amazon Beauty (ABeauty), Amazon Video-Games (AGames), and Amazon Cellphones (ACellphones).[3] We employed the *price* of each item $t$ as

---

[2]Please refer to Appendix D for the related work on the list-wise LTR [5, 27, 37].

[3]http://jmcauley.ucsd.edu/data/amazon/

**Table 2: Comparison of accuracy (*i.e.*, *P*@10 and *G*@10) and profit (*TP*@10 and *PH*@10) with the state-of-the-art methods, where the underlined values denote the best results for the corresponding metrics among those of competing methods, and the bold values indicate the results from our approach (*i.e.*, 'Ours')**

| Dataset | Methods | MF | | | | CDAE | | | | LightGCN | | | |
|---|---|---|---|---|---|---|---|---|---|---|---|---|---|
| | | *P*@10 | *G*@10 | *TP*@10 | *PH*@10 | *P*@10 | *G*@10 | *TP*@10 | *PH*@10 | *P*@10 | *G*@10 | *TP*@10 | *PH*@10 |
| ABeauty | Baseline | 0.0091 | 0.0362 | 27.2130 | 1.1289 | 0.0094 | 0.0370 | 28.3342 | 1.1734 | 0.0095 | 0.0387 | 30.2029 | 1.2530 |
| | Spons-Rec | 0.0091 | 0.0364 | 29.0829 | 1.2041 | 0.0093 | 0.0367 | 29.8063 | 1.2227 | 0.0095 | 0.0383 | 30.4253 | 1.2585 |
| | CPP-PPR | 0.0090 | 0.0359 | 31.4533 | 1.2927 | 0.0093 | 0.0368 | 30.6693 | 1.2607 | 0.0094 | 0.0384 | 30.4020 | 1.2579 |
| | Rec-RL | 0.0089 | 0.0357 | 31.6515 | 1.3009 | 0.0089 | 0.0355 | 32.4254 | 1.3354 | 0.0094 | 0.0385 | 31.2494 | 1.3037 |
| | PE-LTR | 0.0091 | 0.0361 | 28.9407 | 1.1867 | 0.0093 | 0.0365 | 29.9093 | 1.2301 | 0.0095 | 0.0384 | 30.0188 | 1.2490 |
| | VCF | 0.0089 | 0.0355 | 30.7057 | 1.2737 | 0.0089 | 0.0350 | 29.8239 | 1.2303 | 0.0093 | 0.0378 | 29.5103 | 1.2323 |
| | Ours | **0.0097** | **0.0383** | **34.1057** | **1.3724** | **0.0099** | **0.0391** | **33.3893** | **1.3492** | **0.0099** | **0.0388** | **35.4061** | **1.4164** |
| | Gain (vs. *B*) | 6.3% | 5.9% | 25.3% | 21.6% | 5.3% | 5.7% | 17.8% | 15.0% | 4.2% | 0.3% | 17.2% | 13.0% |
| | Gain (vs. $C_a$) | 6.3% | 5.3% | 17.3% | 14.0% | 6.5% | 6.3% | 8.9% | 7.0% | 5.3% | 0.8% | 13.3% | 8.6% |
| | Gain (vs. $C_p$) | 8.7% | 7.4% | 7.8% | 5.5% | 11.2% | 10.1% | 3.0% | 1.0% | 5.3% | 0.8% | 13.3% | 8.6% |
| Dataset | Methods | MF | | | | CDAE | | | | LightGCN | | | |
| | | *P*@10 | *G*@10 | *TP*@10 | *PH*@10 | *P*@10 | *G*@10 | *TP*@10 | *PH*@10 | *P*@10 | *G*@10 | *TP*@10 | *PH*@10 |
| AGames | Baseline | 0.0114 | 0.0463 | 131.5763 | 5.2719 | 0.0119 | 0.0487 | 138.5077 | 5.5022 | 0.0114 | 0.0472 | 134.6468 | 5.3399 |
| | Spons-Rec | 0.0114 | 0.0459 | 150.5091 | 5.9938 | 0.0118 | 0.0480 | 144.3387 | 5.7626 | 0.0113 | 0.0468 | 149.1348 | 5.8992 |
| | CPP-PPR | 0.0114 | 0.0453 | 148.8356 | 5.9704 | 0.0118 | 0.0479 | 145.1591 | 5.7844 | 0.0114 | 0.0465 | 160.0159 | 6.3123 |
| | Rec-RL | 0.0113 | 0.0451 | 148.4110 | 5.9371 | 0.0116 | 0.0462 | 158.4935 | 6.3237 | 0.0113 | 0.0463 | 165.4070 | 6.5253 |
| | PE-LTR | 0.0114 | 0.0456 | 136.4930 | 5.4532 | 0.0118 | 0.0481 | 143.7404 | 5.7095 | 0.0114 | 0.0471 | 137.9798 | 5.4658 |
| | VCF | 0.0112 | 0.0447 | 151.7474 | 6.1532 | 0.0115 | 0.0473 | 146.3431 | 5.8226 | 0.0112 | 0.0461 | 141.5071 | 5.6752 |
| | Ours | **0.0121** | **0.0491** | **161.2911** | **6.5039** | **0.0127** | **0.0532** | **173.6802** | **6.8724** | **0.0119** | **0.0478** | **181.8963** | **7.2900** |
| | Gain (vs. *B*) | 6.1% | 6.1% | 22.6% | 23.4% | 6.7% | 9.2% | 25.4% | 24.9% | 4.4% | 1.3% | 35.1% | 36.5% |
| | Gain (vs. $C_a$) | 6.1% | 7.0% | 7.2% | 8.5% | 7.6% | 10.6% | 20.8% | 20.4% | 4.4% | 1.5% | 31.8% | 33.4% |
| | Gain (vs. $C_p$) | 8.0% | 9.8% | 6.3% | 5.7% | 9.5% | 15.2% | 9.6% | 8.7% | 5.3% | 3.2% | 10.0% | 11.7% |
| Dataset | Methods | MF | | | | CDAE | | | | LightGCN | | | |
| | | *P*@10 | *G*@10 | *TP*@10 | *PH*@10 | *P*@10 | *G*@10 | *TP*@10 | *PH*@10 | *P*@10 | *G*@10 | *TP*@10 | *PH*@10 |
| ACell-Phones | Baseline | 0.0082 | 0.0390 | 36.5803 | 1.3130 | 0.0083 | 0.0405 | 38.2753 | 1.3726 | 0.0083 | 0.0397 | 35.9431 | 1.2907 |
| | Spons-Rec | 0.0082 | 0.0385 | 42.5779 | 1.5347 | 0.0083 | 0.0396 | 40.6666 | 1.4580 | 0.0082 | 0.0393 | 41.5552 | 1.4958 |
| | CPP-PPR | 0.0082 | 0.0379 | 40.4264 | 1.4521 | 0.0083 | 0.0395 | 39.5736 | 1.4168 | 0.0082 | 0.0391 | 43.4995 | 1.5608 |
| | Rec-RL | 0.0078 | 0.0346 | 50.6962 | 1.8101 | 0.0081 | 0.0390 | 41.5904 | 1.4922 | 0.0082 | 0.0386 | 46.5963 | 1.6697 |
| | PE-LTR | 0.0083 | 0.0389 | 41.4068 | 1.4852 | 0.0083 | 0.0400 | 40.7413 | 1.4596 | 0.0082 | 0.0393 | 36.7651 | 1.3207 |
| | VCF | 0.0080 | 0.0373 | 45.0316 | 1.6140 | 0.0080 | 0.0379 | 42.7839 | 1.5406 | 0.0082 | 0.0398 | 37.5453 | 1.3385 |
| | Ours | **0.0087** | **0.0417** | **45.3588** | **1.6318** | **0.0085** | **0.0415** | **47.3420** | **1.6902** | **0.0085** | **0.0408** | **51.8236** | **1.8555** |
| | Gain (vs. *B*) | 6.1% | 6.9% | 24.0% | 24.2% | 2.4% | 2.5% | 23.7% | 23.1% | 2.4% | 2.8% | 44.2% | 43.8% |
| | Gain (vs. $C_a$) | 4.8% | 7.2% | 9.5% | 9.9% | 2.4% | 3.8% | 16.2% | 15.8% | 3.7% | 2.5% | 38.0% | 38.6% |
| | Gain (vs. $C_p$) | 11.5% | 20.5% | -10.5% | -9.9% | 6.3% | 9.5% | 10.7% | 9.7% | 3.7% | 5.7% | 11.2% | 11.1% |

**Table 3: Statistics of the three real-world datasets.**

| Datasets | # of users | # of items | # of ratings | Sparsity |
|---|---|---|---|---|
| ABeauty | 20,926 | 11,170 | 184,367 | 99.92% |
| AGames | 23,766 | 10,386 | 226,619 | 99.91% |
| ACellphones | 27,094 | 10,090 | 188,725 | 99.93% |

$prof(t)$, following the previous studies [4, 18].[4] Moreover, we utilized the *timestamp* information of when $u$ purchased $t$ to calculate $recency(u, t)$. We kept only those users who used more than five items and the items rated by more than five users. The detailed statistics of datasets are shown in Table 3.

---
[4]This paper focuses on scenarios where the profit is *static*; we will explore handling situations where each item's profit changes *dynamically* [22, 32] in our future work.

**Evaluation protocol.** For each dataset, we converted every user rating for the observed items to a unary value of 1, following [13, 24, 36]. We divided all ratings *chronologically* into 60% for training, 20% for validation, and 20% for test sets (*i.e.*, 6:2:2). We measured the accuracy with *Precision* (*P*@*N*), *Recall* (*R*@*N*), and *Normalized Discounted Cumulative Gain* (NDCG, *G*@*N*) where *N*=10. Moreover, we measured the platform's profits with *Total Profit* (*TP*@*N*) [26, 35], *Profit-at-Hit* (*PH*@*N*) [18], and *NDCG for Profit* (*GP*@*N*) [21]. (Please refer to Appendix B for a formal expression of each metric.)

We compared our approach with the following five state-of-the-art methods: (i) Spons-Rec [23], (ii) CPP-PPR [18], and (iii) Rec-RL [26], which belong to the RBA; (iv) PE-LTR [21] and (v) VCF [4], which belong to the MBA. Note that all existing methods and our approach can be applied to any CF models (*i.e.*, *model agnostic*); we employed MF [19], CDAE [36], and LightGCN [12].

We set $\alpha$ and $\beta$ to 0.1 and 0.5, respectively, to determine $\mathcal{M}_u$ for **(S2)**. For more details on model training, such as optimizer, learning rates, and early stopping conditions, please refer to Appendix C.

## 5.2 Results and Analyses

We conducted extensive experiments to answer the following evaluation questions (EQs):

- **(EQ1)** How much does our approach enhance the accuracy and profit compared to the state-of-the-art methods?
- **(EQ2)** How effective is defining and using CPP for observed items? (*i.e.*, the ablation study on **(S1)**)
- **(EQ3)** How effective is employing positive items classified based on the CPP? (*i.e.*, the ablation study on **(S2)**)
- **(EQ4)** How effective is our model training scheme compared to the CPP-based point-wise or naïve profit-aware list-wise LTR? (*i.e.*, the ablation study on **(S3)**)
- **(EQ5)** How scalable is our approach to the number of users?
- **(EQ6)** How do the accuracy and profit vary depending on $\gamma$?

Due to space limitations, we omitted the results for EQs 4 and 6 across all datasets, models, and metrics in this paper; instead, they can be found in Appendix F. The code of our approach is available at https://anonymous.4open.science/r/cpp-pars-D811/.

**(EQ1).** Table 2 reports the comparison results of five competing methods and our approach on three datasets with three base CF models, where 'Baseline' is trained only with the set $\mathcal{T}_u$ according to Eq. (1). Gain (vs. $B$) represents the percentage of 'Ours' compared to the 'Baseline'. Moreover, Gain (vs. $C_a$) and Gain (vs. $C_p$) denote the percentage achieved by comparing 'Ours' with the two competing methods showing the best accuracy and best profit, respectively, among all competing methods (*i.e.*, underlined results).

We can make the following observations from Table 2. (1) Overall, existing MBAs (*e.g.*, PE-LTR [21] and VCF [4]) tend to *fail to provide high accuracy and high profit* compared to existing RBAs (*e.g.*, Spons-Rec [23] and Rec-RL [26]); except in the case of ACellphones/CDAE as the dataset/model, PE-LTR [21] or VCF [4] is *not observed to outperform the RBA* in terms of both accuracy and profit. (2) Nevertheless, the accuracy of our proposed approach, which belongs to the MBA, *consistently exceeds* that of the RBA while achieving high profits. (3) Note that, in the case of ACellphones/MF as the dataset/model, 'Ours' shows lower $TP@10$ than that of Rec-RL by about 10.5%. However, Rec-RL exhibits *significantly degraded accuracy* by approximately 17% lower than 'Ours' and about 7.4% lower than the 'Baseline' in terms of $G@10$. Using Rec-RL is *not practical* in this case because such inaccurate recommendations can considerably *worsen user satisfaction* with the platform. From these results, we have empirically verified that our approach effectively addresses the issues of existing MBAs, enabling it to achieve both high accuracy and profit.

**(EQ2).** To justify the design choice of CPP and verify the effectiveness of model training by using CPP for observed items $\in \mathcal{T}_u$, we used the following *variants of CPP* as the weight of the loss function in Eq. (4) for the existing MBA (*i.e.*, PE-LTR [21]): (i) $prof$; (ii) $prof\text{-}w\text{-}\hat{Q}'$; (iii) $prof\text{-}w\text{-}recency$; and (iv) $CPP(u, t)$. Note that the variant (i) is equivalent to the profit-based weighting scheme adopted by the existing MBAs.

**Table 4: Comparison of accuracy (*i.e.*, $R@10$) and profit (*i.e.*, $GP@10$) among variants of CPP for the point-wise LTR (MF).**

| Methods | ABeauty | | AGames | | ACellphones | |
|---|---|---|---|---|---|---|
| | $R@10$ | $GP@10$ | $R@10$ | $GP@10$ | $R@10$ | $GP@10$ |
| Baseline | 0.0651 | 0.0349 | 0.0853 | 0.0449 | 0.0717 | 0.0383 |
| $prof$ | 0.0649 | 0.0351 | 0.0844 | 0.0445 | 0.0721 | 0.0383 |
| $prof\text{-}w\text{-}\hat{Q}'$ | 0.0649 | 0.0353 | 0.0858 | 0.0450 | 0.0727 | 0.0388 |
| $prof\text{-}w\text{-}recency$ | 0.0661 | 0.0366 | 0.0861 | 0.0452 | 0.0728 | 0.0389 |
| $CPP(u, t)$ | **0.0668** | **0.0367** | **0.0870** | **0.0461** | **0.0746** | **0.0403** |
| Gain (%) | **2.6** | **5.2** | **2.0** | **2.7** | **4.0** | **5.2** |

**Table 5: Effectiveness of employing positive items classified via CPP in terms of accuracy (*i.e.*, $R@10$) and profit (*i.e.*, $TP@10$ and $PH@10$) for the point-wise LTR (MF).**

| Methods | ABeauty | | | AGames | | |
|---|---|---|---|---|---|---|
| | $R@10$ | $TP@10$ | $PH@10$ | $R@10$ | $TP@10$ | $PH@10$ |
| Baseline | 0.0651 | 27.2130 | 1.1289 | 0.0853 | 131.5763 | 5.2719 |
| $CPP\text{-}for\text{-}\mathcal{T}_u$ | 0.0668 | 30.3870 | 1.2452 | 0.0870 | 141.0801 | 5.6705 |
| $CPP\text{-}for\text{-}\mathcal{M}_u$ | **0.0670** | **30.4518** | **1.2457** | **0.0873** | **146.7890** | **5.9047** |
| Gain (%) | **2.9** | **11.9** | **10.3** | **2.3** | **11.6** | **12.0** |

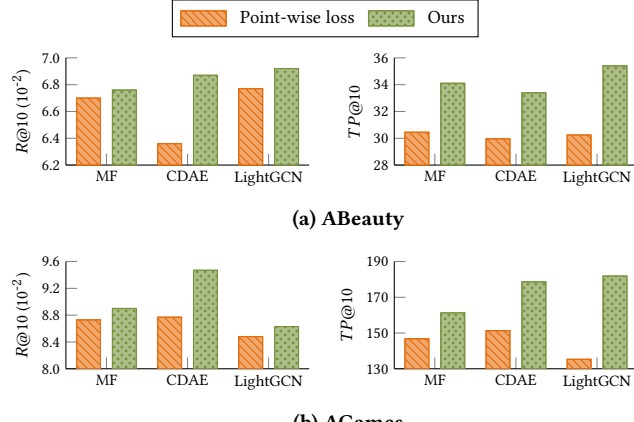

**(a) ABeauty**

**(b) AGames**

**Figure 2: Comparison of accuracy (*i.e.*, $R@10$) and profit (*i.e.*, $TP@10$) between the cases of using the point-wise loss (*i.e.*, cross-entropy) and our proposed CPP-based list-wise loss to train the model $Q$.**

In Table 4, the results are displayed, with 'Gain' representing the percentage of '$CPP(u, t)$' in comparison to 'Baseline (*i.e.*, MF).' It is evident that employing either (ii) $\hat{Q}'(u, t)$ or (iii) $recency(u, t)$ as a weight alongside the item profit leads to enhancements in both accuracy and profit, as opposed to the original PE-LTR using only the profit (*i.e.*, (i) $prof(t)$). Furthermore, the combined use of both (*i.e.*, '$CPP(u, t)$') achieves a *significant improvement* in both accuracy and profit.

**(EQ3).** To show the superiority of using unobserved items that $u$ might prefer as positive items, we compared accuracy and profit

**Table 6: Comparison of accuracy (*i.e.*, $G@10$) and profit (*i.e.*, $GP@10$) among the 'profit-aware' loss functions with MF.**

| Methods | ABeauty | | AGames | | ACellphones | |
|---|---|---|---|---|---|---|
| | $G@10$ | $GP@10$ | $G@10$ | $GP@10$ | $G@10$ | $GP@10$ |
| Point-wise | 0.0354 | 0.0344 | 0.0451 | 0.0441 | 0.0384 | 0.0378 |
| List-wise | 0.0308 | 0.0312 | 0.0394 | 0.0402 | 0.0321 | 0.0323 |
| Ours | **0.0383** | **0.0379** | **0.0491** | **0.0486** | **0.0417** | **0.0414** |
| Gain (%) | 24.4 | 21.5 | 24.6 | 20.9 | 29.9 | 28.2 |

among the following variants, which train the model $Q$ according to Eq. (4): (i) employing $CPP(u, t)$ as the weight for observed items $\in \mathcal{T}_u$ *without using any unobserved items* (*i.e.*, $CPP$-for-$\mathcal{T}_u$); and (ii) employing $CPP(u, t)$ for both the observed items and the positive items classified from unobserved items (*i.e.*, $CPP$-for-$\mathcal{M}_u$). Note that '$CPP$-for-$\mathcal{T}_u$' is equivalent to '$CPP(u, t)$' for **(EQ2)**.

Table 5 shows the results, where MF is used as the baseline and 'Gain' represents the percentage of '$CPP$-for-$\mathcal{M}_u$' compared to 'Baseline.' Across all datasets, '$CPP$-for-$\mathcal{M}_u$' *consistently outperforms* '$CPP$-for-$\mathcal{T}_u$' in terms of both accuracy and profit. These results indicate that identifying unobserved items that $u$ might prefer and *incorporating* them into model training as positive items of $u$ contribute to enhancing both accuracy and profit.

**(EQ4).** To show the effectiveness of our proposed training scheme (*i.e.*, to tackle **(I2)**), we compared 'Ours' with the following method: training $Q$ by the point-wise LTR (*i.e.*, *cross-entropy loss*) employing $CPP(u, t)$ as the weight for each item $t \in \mathcal{M}_u$ (*i.e.*, the same as '$CPP$-for-$\mathcal{M}_u$' for **(EQ3)**).

In Figure 2, regardless of the recommendation models and datasets, our proposed model training scheme *consistently outperforms* the competing scheme in terms of both accuracy and profit. Notably, when using CDAE as the model on the ABeauty dataset, accuracy and profit are enhanced by approximately 8% and 11%, respectively. These results confirm the advantage of training the model to be *directly optimized for item ranking* in profit-aware recommendations.

Moreover, Table 6 shows the results of our proposed scheme (*i.e.*, 'Ours') compared with two naïve versions of the *profit-aware point-wise loss* and the *profit-aware list-wise loss*, which simply use item profit (*i.e.*, $h(prof(t))$) as a weight for each positive item $\in \mathcal{M}_u$. Here, 'Gain' represents the improvement achieved by 'Ours' compared to the naïve profit-aware list-wise loss. For all metrics, the naïve profit-aware list-wise loss is *inferior to* the naïve profit-aware point-wise loss. These results indicate that, when the profit is employed as the weight in a naïve way, it causes the positive items to be *ranked incorrectly*; this makes the naïve profit-aware list-wise loss provide rather worse accuracy and profit compared to the naïve profit-aware point-wise loss. On the other hand, 'Ours' shows *significantly improved* results against not only the naïve profit-aware list-wise loss but also the naïve profit-aware point-wise loss. It verifies that we effectively enhance the original list-wise loss [37] by *incorporating the concept of CPP*, which becomes *more suitable* for profit-aware recommendations.

**(EQ5).** To demonstrate the scalability of our proposed approach, we investigated the change in the training time by varying the total number of users (*i.e.*, $|\mathcal{U}|$). Specifically, we conducted model

**Table 7: Change in training time of our approach according to the total number of users (ACellphones).**

| Elapsed time for the model (sec.) | Total number of users | | | | |
|---|---|---|---|---|---|
| | 5,000 | 10,000 | 15,000 | 20,000 | 25,000 |
| MF | **8.85** | **16.69** | **25.74** | **35.67** | **44.79** |
| CDAE | **8.55** | **17.98** | **28.64** | **40.28** | **53.27** |
| LightGCN | **5.97** | **14.50** | **24.55** | **38.60** | **61.20** |

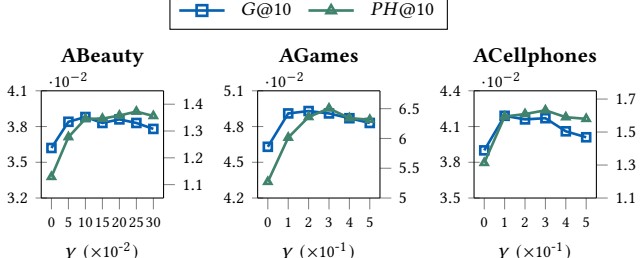

**Figure 3: $G@10$ and $PH@10$ by varying $\gamma$ (MF).**

training by utilizing the interactions of items belonging to randomly sampled user groups, where the sampling percentage varied from 20% to 100% of the total users in an increment of 20%. Then, we observed the *average training time* required for an epoch during this process for each user group.

Table 7 shows the results, where we employed the ACellphones since it has the largest number of users among the three datasets. The elapsed time for the training model *linearly increases* with the increased number of total users, which verifies the time complexity analysis presented in Section 4.5.

**(EQ6).** We show the changes in accuracy and profit with different values for parameter $\gamma$, which is used to adjust how much $\mathcal{L}_u^{prof}$ is reflected in the final loss in Eq. (11).

Figure 3 displays the results, where the $x$-axis denotes $\gamma$, and the $y$-axis denotes the results from the corresponding metrics. The results showing high accuracy and high profit can be obtained when $\gamma$ is set to 0.25, 0.30, and 0.30 on ABeauty, AGames, and ACellphones datasets, respectively. We leveraged these values of $\gamma$ for each dataset in the experiments for previous EQs.

## 6 Conclusions

In this paper, we revealed the following issues of existing MBAs while training the model for profit enhancement: **(I1)** inaccurate inference for the item ranking; and **(I2)** model training that is not directly optimized for the item ranking. To address these issues, we proposed a novel approach that can effectively train the model via learning item rankings obtained from considering both user preferences and item profits. Our approach consists of the following three steps: **(S1)** definition of CPP; **(S2)** classification through CPP; and **(S3)** list-wise LTR based on CPP. Extensive experiments demonstrated that our proposed approach can be more beneficial than existing state-of-the-art methods in achieving high accuracy and high profit.

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

**Table A: Notations**

| Notation | Description |
|---|---|
| $\mathcal{U}$ and $\mathcal{T}$ | A set of all users and items, respectively |
| $CP(u, t)$ | The current preference of $u$ for $t$ |
| $prof(t)$ | The unique profit of $t$ |
| $CPP(u, t)$ | $CP(u, t)$ incorporated with $prof(t)$ |
| $\hat{Q}(u, t)$ | The preference score of $u$ for $t$ predicted by the model $Q$ |
| $Q'$ | The pre-trained recommendation model |
| $\mathcal{T}_u$ (resp. $\overline{\mathcal{T}}_u$) | A set of items observed (resp. unobserved) by $u$ |
| $\mathcal{M}_u$ | A set of positive items of $u$ |
| $\mathcal{N}_u$ | A set of negative items of $u$ |
| $GT_u$ | The ground-truth list for $u$ to compute the list-wise loss |

---

**Algorithm 1** Our proposed approach for training $Q$

---

**Require:** $\alpha$, $\beta$, $\gamma$, and the pre-trained model $Q'$

1: Initialize parameters of $Q$.
2: **for** each user $u \in \mathcal{U}$ **do**
3:   Compute $\mathcal{L}_u^{acc}$ by Eq. (1).
4:   **for** each item $t \in \mathcal{T}_u \cup \overline{\mathcal{T}}_u$ **do**
5:     **if** $t \in \overline{\mathcal{T}}_u$ **then**
6:       $seq(u, t) \leftarrow 1$
7:     $CP(u, t) \leftarrow \hat{Q}'(u, t) \cdot \log(seq(u, t) + 1)$
8:   $\mathcal{H}_u \leftarrow$ Top $\alpha\%$ of items in $\overline{\mathcal{T}}_u$ based on $CP(u, t)$
9:   **for** each item $t \in \mathcal{H}_u$ **do**
10:     $P(t) \propto f(prof(t)) := prof(t)^2$
11:   $\mathcal{M}_u \leftarrow$ Randomly sampled $\beta\%$ of items in $\mathcal{H}_u$ with $P(t)$
12:   $\mathcal{M}_u \leftarrow \mathcal{M}_u \cup \mathcal{T}_u$
13:   **for** each item $t \in \mathcal{M}_u$ **do**
14:     $CPP(u, t) \leftarrow CP(u, t) \cdot h(prof(t))$
15:   $\mathcal{N}_u \leftarrow$ Randomly sampled items in $(\overline{\mathcal{T}}_u - \mathcal{H}_u)$.
16:   **for** each item $t \in \mathcal{N}_u$ **do**
17:     $CPP(u, t) \leftarrow 0$
18:   $GT_u \leftarrow$ A list of items $\in \mathcal{M}_u \cup \mathcal{N}_u$ sorted by $CPP(u, t)$
19:   $R(GT_u | Q) \leftarrow \prod_{i=1}^{|\mathcal{M}_u|} \frac{exp(\hat{Q}(u, GT_u[i]))}{\sum_{k=i}^{|GT_u|} exp(\hat{Q}(u, GT_u[k]))}$
20:   $\mathcal{L}_u^{prof} \leftarrow -\log(R(GT_u | Q))$
21:   $\mathcal{L}_u \leftarrow \mathcal{L}_u^{acc} + \gamma \cdot \mathcal{L}_u^{prof}$
22:   Update $Q$ by the gradient obtained from $\mathcal{L}_u$.
23: Recommend top-$N$ items for $u \in \mathcal{U}$ by the trained $Q$.

## A  Notations

Table A summarizes the notations used in this paper.

## B  Evaluation Metrics

We measured the accuracy with *Precision* ($P@N$), *Recall* ($R@N$), and *Normalized Discounted Cumulative Gain* (NDCG, $G@N$), widely used in existing studies for recommender systems [3, 7, 33, 34], where $N$=10. These three metrics are formulated as follows:

- $P@N = \frac{1}{|\mathcal{U}|} \sum_{u \in \mathcal{U}} \frac{1}{|Rec_u(N)|} \cdot |Rel_u \cap Rec_u(N)|$;

- $R@N = \frac{1}{|\mathcal{U}|} \sum_{u \in \mathcal{U}} \frac{1}{|Rel_u|} \cdot |Rel_u \cap Rec_u(N)|$;

- $G@N = \frac{1}{|\mathcal{U}|} \sum_{u \in \mathcal{U}} \frac{DCG_u@N}{ICDG_u@N}$.

Here, $DCG_u@N$ is formulated as $\sum_{k=1}^N \frac{2^{y_k}-1}{log_2(k+1)}$, where $k$ indicates the rank in $Rec_u(N)$ and $y_k$ represents a binary value that is set to 1 if the $k$th item belongs to $Rel_u$ or set to 0 otherwise. In addition, $IDCG_u@N$ indicates an *ideal DCG* (IDCG), which considers all values of $y_k$ for $Rec_u(N)$ as 1; it serves as the normalizing factor for DCG (*i.e.*, $\frac{DCG_u@N}{ICDG_u@N}$).

Moreover, we measured the platform's profits by the following metrics: *Total Profit* ($TP@N$) [26, 35], *Profit-at-Hit* ($PH@N$) [18], and *NDCG for Profit* ($GP@N$) [21]. These metrics are formulated as follows:

- $TP@N = \sum_{u \in \mathcal{U}} \sum_{t \in Rel_u \cap Rec_u(N)} prof(t)$;

- $PH@N = \frac{1}{|\mathcal{U}|} \sum_{u \in \mathcal{U}} \frac{1}{|Rel_u|} \sum_{t \in Rel_u \cap Rec_u(N)} prof(t)$;

- $GP@N = \frac{1}{|\mathcal{U}|} \sum_{u \in \mathcal{U}} \frac{P\text{-}DCG_u@N}{P\text{-}ICDG_u@N}$

First, $TP@N$ gets improved as a more number of profitable (*i.e.*, ground-truth) items for $u$ are included in the top-$N$ items. Note that, for $TP@N$, we reported the values divided by $10^3$ in the results. Next, $PH@N$ indicates the average profit per user from $Rel_u \cap Rec_u(N)$. Lastly, for $GP@N$, $P\text{-}DCG_u@N$ is formulated as $\sum_{k=1}^N \frac{(2^{y_k}-1) \cdot prof(t)}{log_2(k+1)}$. In addition, $P\text{-}IDCG_u@N$ denotes an ideal DCG obtained by sorting the profits of all items $\in Rel_u \cap Rec_u(N)$ in descending order.

## C  Implementation Details

We set $\alpha$ and $\beta$ to 0.1 and 0.5, respectively, to determine $\mathcal{M}_u$. When using MF or CDAE as base models, we used a value of $4 \cdot |\mathcal{T}_u|$ for the size of negative sampling (*i.e.*, $|\mathcal{N}_u|$) of each user $u$. Moreover, when using LightGCN as a base model, we used a value of $1 \cdot |\mathcal{T}_u|$ for the size of negative sampling. For all competing methods, we fine-tuned the hyperparameters via our grid search on the validation set. Specifically, we adjusted the hyperparameters for each method and reported the results when the decrease in accuracy was within 1% compared to the baseline:

- For Spons-Rec [23], we adjusted $w$ from 0.0005 to 0.006 in an increment of 0.0005.
- For CPP-PPR [18], we adjusted both $\alpha$ and $\beta$ from -0.2 to 1.0 in an increment of 0.1.
- For Rec-RL [26], we employed the average profit of items $\in \mathcal{T}_u$ used by a user $u$ as *state information* in the context of *reinforcement learning*. In addition, we adjusted the noise standard deviation from 0.1 to 0.5 in an increment of 0.1 and the learning rate from {0.0005, 0.001, 0.005, 0.01, 0.05}.
- For PE-LTR [21], we adjusted the constraint term in an interval of 0.1.

Additionally, we employed 256-dimensional embedding vectors for MF, while employing 512-dimensional embedding vectors for CDAE and LightGCN. We set the learning rate as 0.0005, 0.0001, and 0.001 for MF, CDAE, and LightGCN, respectively. We adopted the *adaptive moment estimation* (Adam) optimizer [17] to train these recommendation models, which adapts the learning rate for

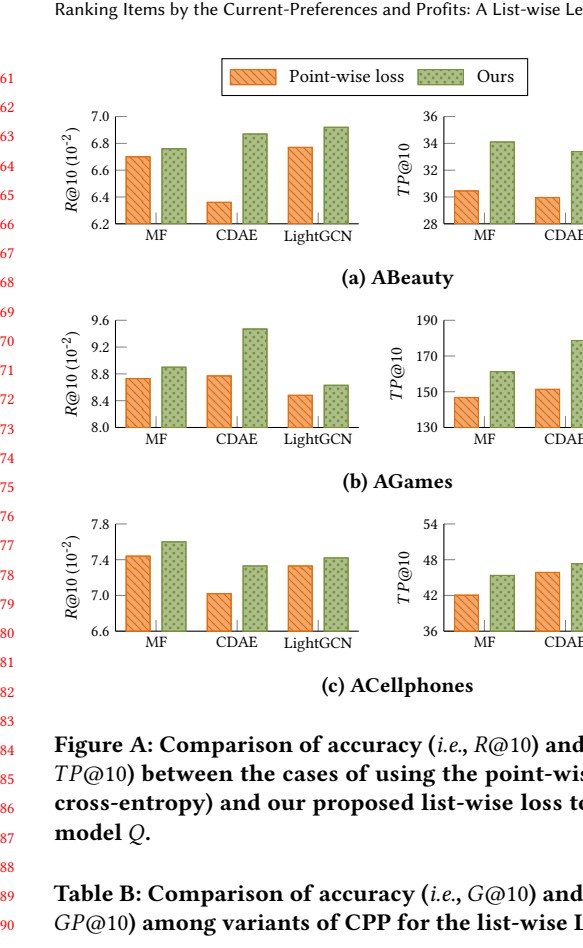

**Figure A: Comparison of accuracy (*i.e.*, R@10) and profit (*i.e.*, TP@10) between the cases of using the point-wise loss (*i.e.*, cross-entropy) and our proposed list-wise loss to train the model $Q$.**

**Table B: Comparison of accuracy (*i.e.*, G@10) and profit (*i.e.*, GP@10) among variants of CPP for the list-wise LTR (MF).**

| Methods | ABeauty | | AGames | | ACellphones | |
|---|---|---|---|---|---|---|
| | G@10 | GP@10 | G@10 | GP@10 | G@10 | GP@10 |
| $prof$ | 0.0308 | 0.0312 | 0.0394 | 0.0402 | 0.0321 | 0.0323 |
| $prof$-$w$-$\hat{Q}'$ | 0.0332 | 0.0332 | 0.0446 | 0.0444 | 0.0367 | 0.0366 |
| $prof$-$w$-$recency$ | 0.0374 | 0.0371 | 0.0468 | 0.0467 | 0.0387 | 0.0386 |
| $CPP(u,t)$ | **0.0383** | **0.0379** | **0.0491** | **0.0486** | **0.0417** | **0.0414** |
| Gain (%) | **2.4** | **2.2** | **4.9** | **4.1** | **7.8** | **7.3** |

embedding vectors. The weight decay values are equivalent to 1e-6, 1e-7, and 1e-4 for MF, CDAE, and LightGCN, respectively. The batch sizes are 512 for MF, 128 for CDAE, and 2048 for LightGCN.

The default maximum number of epochs for all models is set to 500. For MF and CDAE, *early stopping* is triggered if there is no improvement in R@10 on the validation set after 10 checks at 5-epoch intervals. In the case of LightGCN, *early stopping* is applied if there is no improvement in R@10 on the validation set for 50 epochs.

Our experiments were conducted in Linux running on Intel Core i7 (4.0 GHZ) and i9 processors (3.7 GHZ) with Nvidia RTX 2070 and 3070ti. The code has been developed under Python 3.7.13 with the following main packages: PyTorch 1.4.0, NumPy 1.21.6, pandas 0.24.2, and SciPy 1.3.0.

## D Related Work

### D.1 Profit-aware Recommender Systems

Existing studies for profit-aware recommender systems are categorized into the following two groups: (i) *re-ranking-based approach* (*i.e.*, RBA) [18, 23, 26]; (ii) *model-based approach* (*i.e.*, MBA) [4, 21].

Spons-Rec [23] re-ranks items based on a *weighted sum* of $\hat{Q}(u,t)$ and $prof(t)$ according to Eq. (2). CPP-PPR [18] employs the average profit of items used by $u$ with $\hat{Q}(u,t)$ and $prof(t)$; it aims to consider the profit trend of the items frequently used by a user while re-ranking items. Rec-RL [26] employs the *reinforcement-learning* (RL) scheme [16, 31] to combine $\hat{Q}(u,t)$ and $prof(t)$. On the other hand, PE-LTR [21], which belongs to the MBA, trains $Q$ with $\mathcal{L}_u^{acc}$ and $\mathcal{L}_u^{prof}$ aiming to find *Pareto-efficient solutions* [6, 30, 39] according to Eq. (3). VCF [4] trains $Q$ by employing $prof(t)$ as the weight of a loss function according to Eq. (4).

### D.2 List-wise Learning-to-Rank

The list-wise approach [5, 27, 37] is one of the widely adopted *learning-to-rank* (LTR) schemes [5, 20, 24, 25, 27, 28, 37], which trains a model to minimize the *ranking difference* between items in two lists: (i) the predicted list, *i.e.*, a list of items ranked according to scores predicted by the model, and (ii) the ground-truth list.

ListNet [5] computes the cross-entropy between *permutations* obtained from the two lists, and RankCosine [27] computes the *cosine similarity* between the two lists. ListMLE [37] establishes the list-wise loss by formalizing the LTR as the problem of minimizing the likelihood loss between the two lists. Our proposed list-wise loss differs from existing list-wise approaches in that it constructs the ground-truth list by considering the *Current Preference incorporated with Profit* (CPP) for each item to achieve **(G1)** and **(G2)** for a profit-aware recommendation.

## E Pseudocode

Algorithm 1 sketches the process of training the model $Q$ through our approach.

For ease of explanation, we assume the number of epochs for training is 1. In line 3, we compute the loss $\mathcal{L}_u^{acc}$ for $Q$ following the cross-entropy loss to distinguish between items belonging to $\mathcal{T}_u$ and $\overline{\mathcal{T}}_u$. In lines 4–19, we describe obtaining our $\mathcal{L}_u^{prof}$ for the profit enhancement. First, in lines 4–7, we compute $CP(u,t)$, defined by **(S1)**, for each item $t \in \mathcal{T}_u \cup \overline{\mathcal{T}}_u$. Then, in lines 8–12, after finding the set $\mathcal{H}_u$ of unobserved items that $u$ might prefer, we determine the positive items for $u$, according to **(S2)**. In lines 13–14, by combining $CP(u,t)$ and $prof(t)$ for each item $t \in \mathcal{M}_u$, we obtain $CPP(u,t)$. Next, in lines 15–17, we randomly sample negative items for $u$ from $(\overline{\mathcal{T}}_u - \mathcal{H}_u)$ and assign them with $CPP(u,t)$ of 0, according to **(S2)**. In lines 18–20, we compose the ground-truth list $GT_u$ following **(S3)** and then employ $GT_u$ to compute $\mathcal{L}_u^{prof}$. In lines 21–22, by optimizing $\mathcal{L}_u$, we train and update $Q$. Finally, we recommend top-$N$ items through $Q$ trained by all users $\in \mathcal{U}$.

## F Additional Experimental Results

### F.1 Effectiveness of Defining the CPP

To justify the design choice of CPP and verify the effectiveness of model training by using CPP for observed items $\in \mathcal{T}_u$, we used the following *variants of CPP* to determine the *ground-truth list* for the list-wise LTR: (i) $prof$; (ii) $prof$-$w$-$\hat{Q}'$; (iii) $prof$-$w$-$recency$;

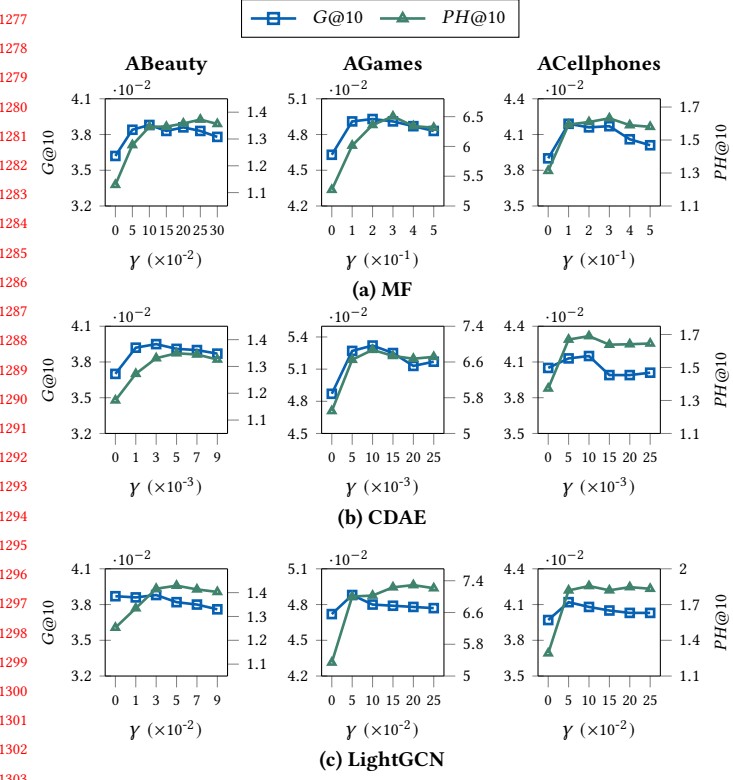

**Figure B: Accuracy (*i.e.*, G@10) and profit (*i.e.*, PH@10) by varying γ.**

**Table C: Comparison of accuracy (*i.e.*, G@10) and profit (*i.e.*, GP@10) among the 'profit-aware' loss functions when CDAE and LightGCN are employed as base CF models.**

| CDAE | ABeauty | | AGames | | ACellphones | |
|---|---|---|---|---|---|---|
| | G@10 | GP@10 | G@10 | GP@10 | G@10 | GP@10 |
| Point-wise | 0.0330 | 0.0323 | 0.0460 | 0.0452 | 0.0392 | 0.0386 |
| List-wise | 0.0276 | 0.0284 | 0.0380 | 0.0392 | 0.0261 | 0.0265 |
| Ours | **0.0391** | **0.0386** | **0.0532** | **0.0530** | **0.0415** | **0.0411** |
| Gain (%) | 41.7 | 35.9 | 40.0 | 35.2 | 59.0 | 55.1 |

| LightGCN | ABeauty | | AGames | | ACellphones | |
|---|---|---|---|---|---|---|
| | G@10 | GP@10 | G@10 | GP@10 | G@10 | GP@10 |
| Point-wise | 0.0378 | 0.0368 | 0.0472 | 0.0458 | 0.0397 | 0.0390 |
| List-wise | 0.0314 | 0.0316 | 0.0340 | 0.0348 | 0.0298 | 0.0299 |
| Ours | **0.0382** | **0.0381** | **0.0478** | **0.0480** | **0.0408** | **0.0407** |
| Gain (%) | 21.7 | 20.6 | 40.6 | 37.9 | 36.9 | 36.1 |

and (iv) $CPP(u, t)$. Note that the variant (iv) is equivalent to our proposed CPP-based list-wise LTR approach.

In Table B, the results are displayed, with 'Gain' representing the percentage of '$CPP(u, t)$' in comparison to 'Baseline (*i.e.*, MF).'

It is evident that employing either (ii) $\hat{Q}'(u, t)$ or (iii) $recency(u, t)$ as a weight alongside the item profit leads to enhancements in both accuracy and profit, compared to using only the profit (*i.e.*, (i) $prof(t)$). Furthermore, the combined use of both (*i.e.*, '$CPP(u, t)$') achieves a *significant improvement* in both accuracy and profit.

## F.2 Effectiveness of CPP-based List-wise LTR

To show the effectiveness of our proposed training scheme, we compared 'Ours' with the following method: training $Q$ by the *point-wise loss* employing $CPP(u, t)$ as the weight for each item $t \in \mathcal{M}_u$ (*i.e.*, the same as '*CPP-for-$\mathcal{M}_u$*' for **(EQ3)**).

In Figure A, regardless of the base recommendation models and datasets, our model training scheme (*i.e.*, 'Ours') *consistently outperforms* the competing scheme based on the point-wise loss in terms of both accuracy and profit. Specifically, when using CDAE as the model on the ACellphones dataset, accuracy and profit are enhanced by approximately 4.4% and 3.2%, respectively. Moreover, when using LightGCN, accuracy and profit are enhanced by approximately 1.2% and 31.9%, respectively. These results validate that our proposed training scheme can more effectively learn the *rankings of items* sorted according to $CPP(u, t)$ than the point-wise loss using cross-entropy, which is primarily employed in existing MBAs.

Furthermore, Table C shows the results of our proposed scheme (*i.e.*, 'Ours') compared with two naïve versions of the *profit-aware point-wise loss* and the *profit-aware list-wise loss*, which simply use item profit (*i.e.*, $h(prof(t))$) as a weight for each positive item $\in \mathcal{M}_u$. As the base CF models, CDAE [36] and LightGCN [12] are employed. Here, 'Gain' represents the improvement achieved by 'Ours' compared to the naïve profit-aware list-wise loss. For all metrics, the naïve profit-aware list-wise loss is *inferior to* the naïve profit-aware point-wise loss. These results indicate that, when the profit is employed as the weight in a naïve way, it causes the positive items to be *ranked incorrectly*; this makes the naïve profit-aware list-wise loss provide rather worse accuracy and profit compared to the naïve profit-aware point-wise loss. On the other hand, 'Ours' shows *significantly improved* results against not only the naïve profit-aware list-wise loss but also the naïve profit-aware point-wise loss. It verifies that we effectively enhance the original list-wise loss [37] by *incorporating the concept of CPP*, which becomes *more suitable* for profit-aware recommendations.

## F.3 Hyperparameter Sensitivity Test for γ

We show the changes in accuracy and profit with different values for parameter γ, which is used to adjust how much $\mathcal{L}_u^{prof}$ is reflected in the final loss in Eq. (11).

Figure B displays the results, where the $x$-axis denotes γ, and the $y$-axis denotes the results from the corresponding metrics. Both accuracy and profit tend to be relatively *insensitive* to γ values above a certain threshold (*e.g.*, γ=0.1 for AGames dataset); The results showing high accuracy and profit can be obtained when γ is set to 0.25, 0.30, and 0.30 on ABeauty, AGames, and ACellphones datasets, respectively. We leveraged these values of γ for each dataset in the experiments for previous EQs.

