# OpenReview forum: "Ranking Items by the Current-Preferences and Profits: A List-wise Learning-to-Rank Approach to Profit Maximization"
_ACM.org/TheWebConf/2025/Conference — WWW 2025 Poster_

### Official Review · Reviewer_37G5 · 2024-11-26

**Novelty:** 3
**Technical Quality:** 3

**Review:**

This paper proposes a new List-wise Learning-to-Rank (LTR) approach, named CPP-based LTR, for profit-aware recommendation systems. The method incorporates Current Preference and Profit (CPP) into the ranking framework, optimizing item recommendations to balance user preferences with platform profitability.

Comments:
1. The baselines chosen for comparison are not representative of the latest advancements.

2. While the CPP-based LTR approach is presented as a generalizable framework, the paper does not demonstrate its feasibility in real-world scenarios. Key practical considerations, such as integration with dynamic pricing or user engagement feedback loops, are not addressed. Without evidence of real-world utility, the framework's practical contributions remain unclear.

3. The paper lacks a discussion of computational efficiency. With an increasing focus on scalability and resource constraints in modern recommendation systems, the absence of training and inference time comparisons against baselines is a significant oversight.

4. CPP is defined as the product of a user’s current preference (CP) and item profit, adjusted by a logarithmic function. While the paper describes CP as incorporating recency and pre-trained preference scores, the specific choice of combining these elements with item profit lacks theoretical justification.

**Questions:**

See the Review.

**Reviewer Confidence:**

3: The reviewer is confident but not certain that the evaluation is correct

**Scope:**

3: The work is somewhat relevant to the Web and to the track, and is of narrow interest to a sub-community

---

### Official Review · Reviewer_P7Gf · 2024-11-26

**Novelty:** 5
**Technical Quality:** 6

**Review:**

This paper proposes a method for Profit-aware Recommendation Systems that maintains accuracy while recommending items with high profitability. According to the paper, the field can be divided into Re-ranking Based Approaches (RBA) and Model-based Approaches (MBA). This paper aims to address the limitations of existing MBA models. To overcome the limitations of previous MBA methods, the authors propose: (1) a new concept of Current Preference Profitability (CPP), which integrates the user's current preferences and the profitability of items, (2) classifying unobserved items based on CPP, and (3) optimizing item rankings using listwise learning-to-rank (LTR) based on CPP. CPP is defined by reflecting the attributes and purchase timing recently preferred by the user while incorporating profitability to rank items effectively. Unobserved items are classified as either positive or negative based on their CPP values, with high-profit items receiving greater weight in the process. Through listwise LTR, the model learns to rank items according to their CPP values, improving both recommendation accuracy and profitability.

The paper effectively highlights the limitations of existing MBA methods and provides a detailed explanation of these challenges. It introduces a novel approach to address them. Moreover, the experimental design strongly supports the authors’ claims, and the authors have conducted the ablation studies thoroughly.


However, the paper compares only a limited number of existing models, possibly because this is a relatively new area. Additionally, the use of inconsistent metrics to evaluate profit—alternating between GP, TP, and PH in different instances—raises concerns about the potential cherry-picking of results. The authors are encouraged to verify this is not the case. On a minor note, in Section 4.5 regarding time complexity, the explanation for determining the complexity of processing *all positive items* is unclear and could be elaborated for better understanding.

**Questions:**

1. Is there a specific reason for not adopting a single consistent metric to evaluate profit across all experiments? What is the reason for using GP@10 in Tables 4 and 6, and TP@10 and PH@10 in Table 5?

2. In Section 2.2, the paper states that RBA methods fail to achieve both (G1) and (G2) simultaneously. However, Table 1 shows that the MF model performs better than the baseline in terms of both accuracy and profit. Could you clarify this apparent inconsistency?

3. In Section 4.5, could you provide a more detailed explanation of how the time complexity for determining *all positive items* was derived?

4. In Figure 3, despite an increase in the weight of profit loss with higher gamma values, there does not appear to be a trade-off between accuracy and profit. Could you provide additional explanation or analysis to account for this observation?

**Reviewer Confidence:**

3: The reviewer is confident but not certain that the evaluation is correct

**Scope:**

3: The work is somewhat relevant to the Web and to the track, and is of narrow interest to a sub-community

---

### Official Review · Reviewer_NrEs · 2024-11-27

**Novelty:** 5
**Technical Quality:** 5

**Review:**

In this paper, the authors propose a new model-based approach for the problem of profit-aware recommendation. The authors put forth the *Current Preference incorporated with Profit* (CPP) metric, which considers both the user's current preference and the profit, and further classifies unobserved items based on this CPP metric in order to utilize such previously neglected information. Furthermore, they adopt a list-wise learning to rank approach instead of the commonly used pointwise approach.

**Strengths**:
- The paper proposes the CPP metric to balance the recommendation accuracy as well as maximize the profits gained from making recommendations.
- The CPP metric is further utilized to classify unobserved items, as some (profitable and desirable) items are yet to be discovered by the user.
- The empirical evaluation on three different datasets highlights the feasibility of the proposed framework, as it outperforms other existing model-based approaches as well as re-ranking-based approaches. Furthermore, the proposed approach is model agnostic and can be easily applied to any CF models.
- The paper is relatively well organized and easy to understand.

**Weaknesses**:
- The datasets used in the empirical evaluation are relatively small.
- It is eassentially a two-stage framework, as it requires a pre-trained model to classify the unobserved items.
- The performance of different CF models (i.e. the 'baseline' row) seem to be very similar to one another, in terms of the accuracy metric, for all 3 datasets.

**Questions:**

The performance of different CF models (i.e. the 'baseline' row) seem to be very similar to one another, in terms of the accuracy metric, for all 3 datasets.
- It seems weird that CDAE and LightGCN perform quite similarly to the basic MF model?
- Furthermore, the proposed approach seems to perform rather similarly for the accuracy metrics, regardless of the base model used. The key difference seems to be in terms of the profit metrics. S2 generates additional samples as a form of 'distant supervision', and this should be helpful for the model's accuracy as well?

**Reviewer Confidence:**

3: The reviewer is confident but not certain that the evaluation is correct

**Scope:**

3: The work is somewhat relevant to the Web and to the track, and is of narrow interest to a sub-community

---

### Official Review · Reviewer_Bwrd · 2024-12-01

**Novelty:** 5
**Technical Quality:** 6

**Review:**

**Main idea**:  The authors propose a recommendation algorithm that aims to both show relevant items for a user and increase profit.

### Strengths:
- The authors present a solid algorithm and compare it with baselines, demonstrating its effectiveness in both providing relevant recommendations and increasing profit.
- The experimental results are comprehensive, covering various datasets and evaluation metrics.

### Concerns and Limitations:
- Overall, the work consists of a combination of existing approaches rather than introducing novel techniques, although it does provide performance improvements.
- The authors could provide more justification for their choice of baseline models (MF, CDAE, LightGCN). A brief discussion on why these specific models were selected as baselines would strengthen the experimental setupю
- The formula on line 247 is referred to as cross-entropy, although it is simply a sum of logarithms without labels, which is not entirely accurate.
- The Notations table (line 1045) is missing some variables. It would be beneficial to include additional variables such as Rec and Rel, as they are used in the metrics mentioned nearby.

**Quality**: The quality of the work is good, with a well-designed algorithm, thorough experiments, and clear explanations. However, there are some minor issues with terminology and notation that could be improved.

It is also commendable that the authors have made their code publicly available, promoting reproducibility and enabling further research building upon their work.

**Novelty**: While the work does not introduce groundbreaking new techniques, the combination of existing approaches and the focus on balancing relevance and profit is a valuable contribution.

#### Significance:
Overall, this is a solid work that addresses an important problem in recommendation systems by balancing relevance and profit. While not highly novel, it presents a practical solution and provides insights into this trade-off.

**Questions:**

- Could the authors provide more insight into the rationale behind selecting MF, CDAE, and LightGCN as baseline models for comparison?

- Looking on Figure 3 and Figure B, one might expect that as γ increases, emphasizing profit over relevance, the accuracy metrics would decrease while the profit metrics increase, and vice versa. However, the plots suggest that both types of metrics move in tandem. What could be the reason behind this synchronous behavior, which appears to contradict the expected trade-off between accuracy and profit?

**Reviewer Confidence:**

2: The reviewer is willing to defend the evaluation, but it is likely that the reviewer did not understand parts of the paper

**Scope:**

4: The work is relevant to the Web and to the track, and is of broad interest to the community

---

### Official Review · Reviewer_xxiS · 2024-12-02

**Novelty:** 4
**Technical Quality:** 3

**Review:**

The paper addresses the problem of profit-aware recommendation,
i.e., if two, possibly conflicting goals should be achieved: to
recommends items the user will prefer/buy as well as items
with a high profit (for the seller). The authors propose to add
an auxiliary loss to a recommender model that encourages
the model to predict a set of items that have been selected based
on original predicted preference, regency and profit. In an experiment
on three public, well-known datasets they show that this approach
outperforms several baselines both in accuracy and profit.

The problem of profit-aware recommendation is clearly interesting.

However, the paper has several major weaknesses:
- w1. In the experiments, the proposed method outperforms the baselines
  in both, accuracy and profit.
  - This way, the model does not solve the trade-off between accuracy
  and profit, but it seems to leverage actually item side-information
  (here, the price) or context-information (recency) that seems to
  have a positive impact on accuracy.
  To separate this effect from the profit-awareness one would need
  to use an attribute and/or context-aware baseline model that gets
  at least the price as item attribute and recency as context.

- w2. Some aspects of the proposed method are stated more complicated
  than necessary (or not clear).
  - a. alg. 1, line 18: you say you sort the item list GT, but the loss term then
    does not depend on this sorting: it just depends on choosing the top
   |M_u| elements. It seems easier to say to choose the
   top-|M_u| elements from M_u \cup N_u.
  - b. alg. 1, line 19: what is the difference between "\hat Q" and "Q" ?
    (It sounds like both denote the same thing.)

- w3. The experiments do not reproduce any experiments from the
  model-based profit-aware recommender literature, such as
  [4] and [21]. This way one cannot compare results with published
  results.

**Questions:**

- q1. How does your method perform if you apply it on top of an
  attribute-aware base recommender that has access to the price
  as item attribute?
- q2. How do you explain that your method can improve both,
  accuracy and profit?
- q3. How does your method compare on the datasets used in
  [4] and [21] ?

**Reviewer Confidence:**

3: The reviewer is confident but not certain that the evaluation is correct

**Scope:**

4: The work is relevant to the Web and to the track, and is of broad interest to the community